# The Influence of the Physiotherapeutic Program on Selected Static and Dynamic Foot Indicators and the Balance of Elderly Women Depending on the Ground Stability

**DOI:** 10.3390/ijerph18094660

**Published:** 2021-04-27

**Authors:** Joanna Matla, Katarzyna Filar-Mierzwa, Anna Ścisłowska-Czarnecka, Agnieszka Jankowicz-Szymańska, Aneta Bac

**Affiliations:** 1Faculty of Motor Rehabilitation, The Bronislaw Czech University of Physical Education, Jana Pawła II 78, 31-571 Krakow, Poland; joanna.kus91@gmail.com (J.M.); katarzyna.filar@awf.krakow.pl (K.F.-M.); anna.scislowska@awf.krakow.pl (A.Ś.-C.); 2Faculty of Health Science, University of Applied Science in Tarnow, ul. Mickiewicza 8, 33-100 Tarnow, Poland; jankowiczszymanska@gmail.com

**Keywords:** seniors, physical activity, stable ground, unstable ground

## Abstract

Seniors are a constantly growing group of people in many societies. It is necessary to develop physiotherapeutic programs to improve their mobility. The aim of this study was to assess the impact of the physiotherapeutic program conducted unstable ground on selected indicators of motor functions of elderly women. Sixty women (60–80 years) participated in the research. Group A (*N* = 20) underwent a 12-week physiotherapeutic program on stable ground, group B (*N* = 20) followed an exercise program on unstable ground, and group C (*N* = 20) (control group) had no therapeutic intervention. The effects of the therapy were assessed by using a FreeMed platform (foot load analysis) and a Biosway balance system. The results were compared using ANOVA (the one-way analysis), the Kruskal–Wallis test and also the post hoc tests (Tukey’s test and the multiple comparison test). In group A, a statistically significant change was observed in the static test and balance assessment, in group B this was observed in the static and dynamic foot tests and balance assessment, in group C, no statistical significance was achieved. The authors’ physiotherapeutic program had a statistically significant effect on changes in the balance and selected indicators of the motor functions of the examined people. Comparing the results before and after the therapy more improvement changes were noted in women training on an unstable ground compared to women training on a stable ground.

## 1. Introduction

The effects of ageing populations and the related consequences are the subject of research in many scientific disciplines [1]. Physiotherapy in geriatrics is a separate field dealing with the influence of physical activity on the physical and mental health of the elderly. Physiotherapeutic programs for seniors allow them to reduce functional limitations, improve motor skills, and improve the quality of life. Therefore, it is necessary to promote physical activity among the elderly in order to ensure a dignified and independent old age [2].

The sedentary lifestyle in a society where the average lifespan is systematically increasing is conducive to the occurrence of civilization diseases. Taking part in regular physical activities has a positive effect on the elderly by improving their mood, health condition, increasing their independence for everyday functioning, and in this way, improves the quality of their lives [3].

The issue of the physical activity of seniors is a reason to search for an optimal training program, ensuring physical independence on a daily basis, despite the passage of years [4,5]. Institutional units, such as the World Health Organization, the Council of Europe, and the national policy of the State, implement a number of recommendations regarding the participation of seniors in physical activity. The proposed recommendations differ and do not clearly indicate which forms of movement are best for seniors [6,7,8].

When creating a training program for seniors, a physiotherapist should focus on the basic forms of movement (e.g., reaching for an object, picking up objects, standing on one leg, walking, etc.) which positively influence the psychophysical health of the mentees. It is recommended that every physiotherapeutic program used in geriatrics should include exercises improving cardiovascular and respiratory efficiency (endurance exercises), as well as resistance exercises involving the large muscle groups. Another component is exercises improving flexibility, balance, and agility to prevent falls in the elderly. At the conclusion of an exercise session, stretching exercises are recommended to allow the body to cool down after training and make it more flexible [9].

In physiotherapy, elements of ground instability are introduced in order to increase the efficiency of classic exercises after an injury or as a supplement to training. Devices such as bosu balls, gymnastic balls, and sensory berets are most often used for this purpose. In the case of seniors, the above-mentioned elements of ground instability should be introduced to improve balance, which deteriorates with age, but they are not used throughout the exercise cycle [10].

Currently, researchers pay a lot of attention to the training elements that should be met to make the physical activity dedicated to seniors effective and safe. These include frequency, duration, number of repetitions, type of exercise, position, and additional equipment, among others [11,12]. However, the available literature on the subject does not deal with the role of unstable ground in exercise programs intended for seniors.

To the best of our knowledge, our study is the first to analyze the effectiveness of the authors’ rehabilitation program conducted on stable and unstable ground in comparison to the control group, so our findings complements the current knowledge in the field of physical activity intended for seniors. Consequently, the aim of the study was to assess the impact of the physiotherapeutic program on stable ground in comparison to physiotherapeutic program on unstable ground on selected static and dynamic foot indicators and balance of elderly women. We hypothesized that exercises on unstable ground have a greater impact on the improvement of selected static and dynamic foot indicators and balance than exercises on a stable ground in elderly women.

## 2. Materials and Methods

### 2.1. Participants

Sixty women qualified for the study (see Figure 1). The participants were assigned randomly by the first author to an experimental groups and a control group. Qualification was based on simple randomization (a coin toss) and was carried out by the main author. Patients assigned to the three groups of 20 patients each: first experimental group (A) underwent a rehabilitation protocol that included exercises on the stable surface, the second experimental group (B) underwent a rehabilitation protocol that included exercises on the unstable surface. The third control group (C) had no intervention.

The inclusion criteria for the study were as follows: age between 60 and 80 years; consent for participation in the study; no contraindications to physical activity; a mini-mental state examination (MMSE) score of 27 or more (no cognitive impairment or dementia); and (experimental groups only) a declaration of consistent participation in the rehabilitation program. The exclusion criteria included: motor disability that prevents independent activity; neurological diseases that cause balance and gait disorders; physical conditions that prevent participation, such as severe respiratory or circulatory deficiency; contraindications to physical activity; an MMSE score of 26 or less (cognitive impairment or dementia); discontinuation of the physiotherapy program during its duration; and participation in another rehabilitation program.

The examined women were in age from 60 to 80 years. The detailed characteristics of the examined women are presented in Table 1.

The research project has been approved by the Bioethics Committee as study No. 126/KBL/OIL/2018. Furthermore, all women who applied to participate in the study were informed about the medical considerations and provided informed consent for participation.

### 2.2. Methods

In order to determine the basic morphological indicators and the inclusion/exclusion criteria, a single measurement of body weight and height was performed, and the level of cognitive functions was determined using the MMSE scale for each woman participating in the project.

All participants in the study were subjected to two assessments of the selected indicators of motor functions: in the case of groups A and B, the first measurement took place before the start of the physiotherapeutic program and the second after its completion (12 weeks later). In control group C, two measurements were also made and the time interval between them was 12 weeks.

Body height and weight test—body height was measured using a Martin-type anthropometer (Seritex, New York, NY, USA) with an accuracy of 0.1 cm. The patient’s height was measured from the top of the head (vertex) in the horizontal plane to the plantar plane of the feet (basis). Body weight was determined using a Tanita scale (Tanita Corporation, Tokyo, Japan) with an accuracy of 0.1 kg.

Cognitive ability test—the MMSE is a 30-item screening tool used to assess cognition including orientation, attention, language, and memory. A patient receives a 0 or 1 score for each question. The interpretation of the scale is as follows: scores of 30–27, normal; 26–24, cognitive impairment; 23–19, mild dementia; 18–11, moderate dementia; 10–0, severe dementia [13].

Static and dynamic foot load test—a comprehensive load analysis was performed using a FreeMED platform (Sensor Medica, Rome, Italy), consisting of an active panel of 40 × 40 cm^2^ (0.16 m^2^) which contained recording sensors, and additional passive panels of 2 × 100 cm^2^, constituting an extension of the active panel [14].

During the static tests on the FreeMed ground reaction force platform, the foot load distribution was calculated. Each static test was performed in a free-standing position, with the arms hanging freely beside the torso and feet parallel to each other, slightly apart, and barefoot. The first measurement was a test measurement and the second was the main measurement. The results of the tests included: individually for the right and left feet—load (kg), maximum load (gr/cm^2^), average load (gr/cm^2^), load by regions—forefoot and hindfoot (%).

During the dynamic tests on the FreeMed ground reaction force platform, selected foot indicators during walking were calculated. During each dynamic test, the patient was asked to walk on the measurement path 12 times at their own pace. Before the actual measurement began, the patient walked on the path five times to prepare for the study. The results of the measurements included: foot area [cm^2^], maximum foot load (gr/cm^2^), average foot load (gr/cm^2^), foot load by region—forefoot and hindfoot (%), and duration of the gait cycle taking into account the single and double support phases (ms).

Balance test—the Biodex platform (Biosway, New York, NY, USA) is a device equipped with an appropriately configured platform and monitor. It also includes a foam covering to imitate unstable ground during one of the available tests.

Biosway can help to assess balance, as well as improve it through the included training programs [15]. The platform offers three standardized testing protocols and six interactive modes of training to match different problems and levels of fitness. The device provides a repeatable and objectively reliable assessment of neuromuscular control and balance on both stable and unstable ground. It can also help to evaluate treatment progress and document the rehabilitation of patients with balance disorders. The device is simple and convenient to use and is particularly recommended for assessing fall risk among seniors and in post-amputation rehabilitation, post-injury orthopedic rehabilitation, sports medicine programs, neuromuscular control disorders, and screening before and after head injuries [15].

During the study on the Biosway platform three test were performed:

Postural Stability Test (PST)—which emphasizes a patient’s ability to maintain center of balance (30 s). Under physiological conditions, information from the vestibular system, the organ of vision and deep sensation receptors, located in muscles, joints, and skin, they enable the correct orientation of the body in space and maintain the center of gravity (COG). This applies to both position relative to the support surface and the limits of stability. The theoretical limit of human posture stability is the surface, passing through the center of gravity, whose shifts can cause a fall. Under normal conditions, the balance control system provides a certain margin of stability, i.e., the optimal center of gravity position relative to the limits of stability. The patient’s score on this test assesses deviations from center, thus a lower score is more desirable than a higher score. The appropriate static scale was selected based on each participant’s body height. Patient height is entered so that the patient’s center of gravity (COG) can be estimated 55% of the patient’s height is used to calculate the COG. Based on the COG height, the BioSway takes into account that the theoretical angular excursion of the COG is different for different height people [16,17].

Limits of Stability—this test assesses a participant’s ability to maintain his or her center of pressure outside the plane of support. The LOS for balance in a standing position are determined based on the maximum angle at which a participant is able to tilt away from the vertical position without losing balance. When LOS are exceeded, the participant either falls or must engage in corrective strategies to prevent a fall, such as taking a step with one leg or bending his or her knees. The test involved mobilizing the participant, to move and control her center of pressure within the base of support while keeping her feet on the ground. During each assessment, nine points were displayed on a screen. The participant was asked to look at the monitor and shift her body weight to make the cursor move from the center of the screen to a flashing point and back again as fast as possible [18].

Modified Clinical Test of Sensory Interaction and Balance (m-CTSIB)—the third test performed on the Biosway platform was m-CTSIB, which assesses fall risk. The test is a reliable method for assessing balance disorders and can also identify disorders related to various systems engaged in postural control, i.e., visual control and the vestibular and somatosensory systems. The results of the test correlate strongly with fall risk. In this study, m-CTSIB was conducted once per participant. The test started after the participant took her position on the platform and was informed about the course of the test, and the positions of her feet were obtained (feet parallel). The test consisted of four parts, each lasting 30 s. In the first part, participants held on a solid surface to assess their visual, vestibular, and somatosensory control. In the second part, participants held a standing position with their eyes closed to assess their vestibular and somatosensory control. The third part involved holding a static position while standing on a dynamic surface with eyes open to assess the interaction between the visual and somatosensory systems. The fourth part was performed in the same manner as the third, except with eyes closed, to assess the interaction between the somatosensory and vestibular systems [19].

### 2.3. The Authors’ Rehabilitation Program

The rehabilitation program lasted 12 weeks and consisted of two 45-min sessions per week. The program was identical for both experimental groups and the only difference was the ground—stable for group A (hard floor); unstable for group B (10 cm thick rehabilitation mattress). During the training, no additional equipment was used and all exercises were based on the use of their own body weight. Each session was planned separately, exercises in the sessions were not repeated and the senior women performed 25–30 rehabilitation exercises during each session.

Each session consisted of three phases and during each phase all neck, torso, and upper and lower limbs muscle groups were involved:

The first phase (approximately ten minutes) involved standing position exercises aimed at preparing the body for effort in the main part of the training and protecting it against possible injuries. Exercises in this phase consisted of slow circular movements, starting with the head, through the upper limbs, the torso and ending with the lower limbs and also marching, clapping, and stamping.

The main phase (approximately 25 min) involved lying, sitting, and standing position exercises. In each position, the participants performed strength, balance, and coordination exercises to achieve better functional fitness and improve their well-being. Each session included a variety of exercises moving all muscle groups. Participants performed ten repetitions of each symmetrical exercise and five repetitions of each asymmetric exercise per side.

The last phase (approximately ten minutes) included lying, sitting, and standing position exercises based on static stretching and respiratory exercises, the purpose of which was to cool down after effort. Participants held each stretched position for 30 s as they progressed through successive groups of muscles, starting from the head, through the torso, towards the distal parts of the limbs.

### 2.4. Statistical Methods

Statistical analysis of the collected data was performed using Statistica v 13 software (StatSoft, Hamburg, Germany). The consistency of distributions with the normal distribution was verified with the Shapiro–Wilk test and the homogeneity of variance was assessed with the Lavine F test. Descriptive statistics were calculated: arithmetic mean, standard deviation, median, as well as the minimum and maximum values. To evaluate differences in the results obtained before and after the applied therapy, the Student’s *t*-test for dependent variables and, alternatively, the Wilcoxon signed-rank test were used. The level of statistical significance was *p* < 0.05. The results obtained in the three studied groups were compared using the one-way analysis of variance (ANOVA) and, alternatively, the Kruskal–Wallis test. If the results obtained in the three groups were not equal, the analysis was continued with the post hoc test, which was Tukey’s test in the case of the parametric test, and the multiple comparison test in the case of the non-parametric test.

## 3. Results

Comparing the results of the selected indicators of the static test for the left foot, statistically significant changes in the average load (gr/cm^2^) were observed in both studied groups. In group C, none of the analyzed indicators of the static test were statistically significant in relation to the time before and after therapy. As for intergroup comparisons, a statistically significant change was observed in the forefoot load (%) between the group exercising on unstable ground (B) and the control group (C) before the applied therapy (see Table 2).

In the case of the right foot, statistically significant differences for selected indicators of the static test were observed only in relation to the average load (gr/cm^2^) in the group exercising on unstable ground (B). In both group A (exercising on stable ground) and group C (control group), none of the analyzed indicators of the static test were statistically significant in relation to the time before and after therapy. As for intergroup comparisons, statistically significant differences were observed in the load (kg) between groups A and C, both before and after the applied therapy (see Table 3).

Comparing the results of the selected indicators of the dynamic test for the left foot, statistically significant changes in the area (cm^2^), forefoot and hindfoot loads (%) as well as single and double support (ms) and gait cycle (ms) were only observed in the group exercising on unstable ground (B). In both group A (exercising on stable ground) and group C, none of the analyzed indicators of the dynamic test for the left foot were statistically significant in relation to the time before and after therapy. As regards intergroup comparisons, statistically significant differences were observed in the maximum load (gr/cm^2^) between groups A and C after the applied therapy, in the average load (gr/cm^2^) between groups A and C both before and after therapy, and between groups A and B after therapy. The last statistically significant change between the groups was observed in double support (ms) between groups A and B before therapy (see Table 4).

Comparing the results of the selected indicators of the dynamic test for the right foot, statistically significant changes were observed in all indicators in the group exercising on unstable ground (B). In group A, statistically significant changes were only observed in forefoot and hindfoot loads (%). No statistically significant changes were observed in group C. As regards intergroup comparisons, statistically significant differences were only found in the average load (gr/cm^2^) after therapy between groups A and B as well as A and C (see Table 5).

In the balance test, the postural stability test (PST) results obtained for the individual groups did not differ significantly between two consecutive measurements in any of the studied groups. The limits of stability (LOS) only differed statistically significantly in the measurements before and after therapy in group B (*p* = 0.005). In the case of intergroup comparisons, they only differed after therapy between groups A and C as well as B and C.

In the modified clinical test of sensory interaction and balance (CTSIB), statistically significant results in two consecutive measurements were obtained in the test with the eyes open on hard ground in group B and in the test with the eyes open on soft ground in groups A and B. As for intergroup comparisons in this test, significant differences were obtained in the test with the eyes open on hard ground between groups B and C before therapy and in the test with the eyes open on soft ground between groups A and C after therapy (see Table 6).

## 4. Discussion

The static test for the musculoskeletal system is a valuable diagnostic tool that can be used to check the quantitative load indicators of the lower limbs in the elderly. The static function of the foot enables proper support and balance of the body in a spatial position [20]. According to Parcou [21], involutional changes in the feet are more common in women and take the form of deformities of the toe and metatarsal bones, translating into static-dynamic failure of the plantar surface of the foot with progressive overload of the forefoot. The study conducted by Puszczałowska-Lizis [22] on foot loads in geriatric women has shown that involutional changes progress to a similar degree in the right and left feet. However, there are no reports in the literature on the influence of physical activity on changes in foot load in the statics of elderly people. In our study, these changes were analyzed in relation to the type of ground and the therapy used. It was shown that the applied therapy only improved the scope of the average load on both feet in people training on unstable ground and improved the same index in the left foot in people training on stable ground.

Dynamic gait analysis in adults, measured by the pressure of the feet on the ground, provides valuable information on existing disorders, and is helpful in the further diagnosis and treatment of abnormalities [23]. After the age of 65, the process of gait automation deteriorates due to changes in the functioning of the nervous system. The feedback synchronization between the work of the skeletal muscles and the central nervous system is weakened, leading to an imbalance in gait [24]. Takehiko et al. [25] have presented prospective studies on changes in selected gait indicators and the risk of disability in people over 65 years of age. They have shown that as the gait cycle lengthens, the level of disability increases in the following years. They have also confirmed the beneficial effect of physical activity on the improvement of gait time indicators in the elderly. Kwon et al. [26] have assessed the risk of falling in elderly people based on dynamic gait analysis. The researchers have compared the gait pattern of people who have fallen and those who have never fallen. They have shown that in people after a fall, the duration of the double-support phase is extended in favor of shortening the swing phase with a simultaneous increase in the maximum load on the supporting foot. However, there are no reports in the available literature on the influence of physical activity on changes in foot load during walking in the elderly in relation to ground stability. Our study completes this gap, pointing to statistically significant differences in the duration of the single- and double-support phases in the group training on unstable ground (B) for both feet. In each of the phases mentioned in this group, the duration of the phases was shortened. In the remaining groups, these indicators did not improve significantly.

Foot load during walking is another dynamic indicator that should be considered for the elderly. Hessert et al. [27] examined the forces of pressure on the ground in elderly people. During the foot transfer phase, they observed a significant load on the lateral edge of the foot, which resulted in imbalance due to the reduced rebound force. Gimunova et al. [28] compared foot load during walking, taking account of sex and age groups: 60–69 years and 70–79 years. Among women, regardless of age, they found significantly higher foot load values in the forefoot area than in that of the hindfoot. However, as in the case of static foot loading, in the field of dynamic foot loading, the available literature also lacks reports analyzing changes in these indicators under the influence of a therapeutic program in relation to ground stability. Our study shows such dependencies. The most statistically significant changes in dynamic indicators were noted in the group training on unstable ground (B) in relation to the other groups. A greater percentage of these changes concerned the right foot. In the group exercising on stable ground, a statistically significant change was only observed in the forefoot load distribution in relation to the hindfoot in the right foot, while in the control group no statistically significant changes were observed.

In the case of the elderly, special attention should be paid to the balance control system. It determines the spatial position of the general center of gravity of the body with the help of the eyes, the vestibular system, and proprioceptors [29]. An interesting study was presented by Sample et al. [30] who analyzed the results of anterior–posterior and medial–lateral deflections in two groups of seniors (150 people). The researchers examined how the type of activity undertaken influenced the results of balance measurements in people who had fallen within the previous 12 months and those who had not had such an incident. The results confirmed greater balance control in people who had not yet fallen. They also showed an interesting dependence of the reduction in the level of imbalance in the case of combining motor and cognitive activities. Maixnerova et al. [31] assessed balance in seniors training with the use of the Nintendo Wii system and compared the results with those who did not train. Importantly, they paid attention to how eyesight was used to control balance. Improvement in the results was noted for both open and closed eyes, with a medium change for open eyes, and small for closed eyes. The control group, without therapeutic intervention, did not obtain a significant change in the assessment of balance compared to the training group. Pluchino et al. [32] assessed the effects in the elderly of eight weeks of Tai Chi training and home balance training with video games. The researchers did not find any significant differences in balance control between the studied groups and identified the anterior–posterior direction as the most disturbed in balance control in seniors. In our study, a statistically significant improvement was only obtained in relation to the movement direction control indicator, and only in the group exercising on unstable ground (B). In our study, the measurement of balance in relation to the eyes showed a significant statistical improvement in the test on hard ground with the eyes open for people training on unstable ground (B) and on soft ground for people training on stable or unstable ground (A and B). As regards the measurement with the eyes closed, no statistically significant differences were found in any of the groups. In the control group (C), no statistically significant changes were observed for any of the indicators tested.

A properly selected form of movement increases interest in exercise and encourages the elderly to continue their activity. Mobily [33] conducted a cohort study on a group of 1103 seniors to see what guided the elderly when choosing exercises and what factor determined their continuation. People who maintained physical activity at the level of five sessions a week after the end of the study chose moderate-intensity exercises, taking several forms of movement into account. It turned out to be important to perform the same activity in different spaces. Josephsons et al. [34] took into account the forms of physical activity intended for seniors and assessed the level of change in the studied groups depending on the type of exercise. People participating in the project were assigned to three groups: the first performed traditional strength training without equipment, the second underwent training enriched with elements of functional training using additional accessories, and the third was a control group who did not train. The researchers showed that the greater improvement in the assessment of the measured indicators concerned the group with elements of functional training using additional equipment. Souza et al. [35] noted the benefits of physical activity in the elderly. They emphasized the importance of changes in the psychophysical indicators under the influence of exercise in relation to people of the same age. The researchers proposed assessing the effects of therapy at individual stages of ageing due to discrepancies arising in the overall assessment of the entire population of people over 60 years of age.

Our findings supported the hypothesis that exercises on unstable ground have a greater impact on the improvement of selected static and dynamic foot indicators and balance than exercises on a stable ground in elderly women. Obtaining a significant improvement in some balance indicators and static indicators and the lack of any statistically significant changes in the control group confirm that the physical activity of the elderly has a positive effect on improvements in their physical performance, and the additional element, which is unstable ground, makes it possible to achieve even better results.

The beneficial influence of physical activity on the elderly is indisputable; however, the question of what the optimal training program is remains open. It is crucial to develop exercises that will encourage seniors to participate in training and achieve satisfactory results at the psychomotor level, which will translate into an improvement in the quality of life.

## 5. Study Limitation

This study is not without limitations. Increasing the frequency of training would be good guidelines for the future studies. Furthermore, all participants of this study lived in a large city, meaning that women from small cities or villages were not included. This is important due to the fact that a rural lifestyle is slightly different to an urban lifestyle, and that a larger-scale study could bring a broader insight into the subject matter. Moreover, sample size was technically and organizationally limited: access to people who have responded positively to the invitation to participate in the research and met the inclusion criteria; limited funds that the authors of the project could allocate for research (the project was entirely financed from the authors’ own funds); lack of literature data that would make it possible to calculate the required sample size at the research planning stage.

## 6. Conclusions

The authors’ physiotherapeutic program had a significant impact on changes in selected indicators of the motor functions of the examined people., Comparing the results before and after the therapy more improvement changes were noted in women training on an unstable ground compared to women training on a stable ground, particularly with regard to dynamic foot loading. The practical implication of our research is the advice for practicing physiotherapists to introduce exercises on an unstable surface when creating exercise programs for older women.

## Figures and Tables

**Figure 1 ijerph-18-04660-f001:**
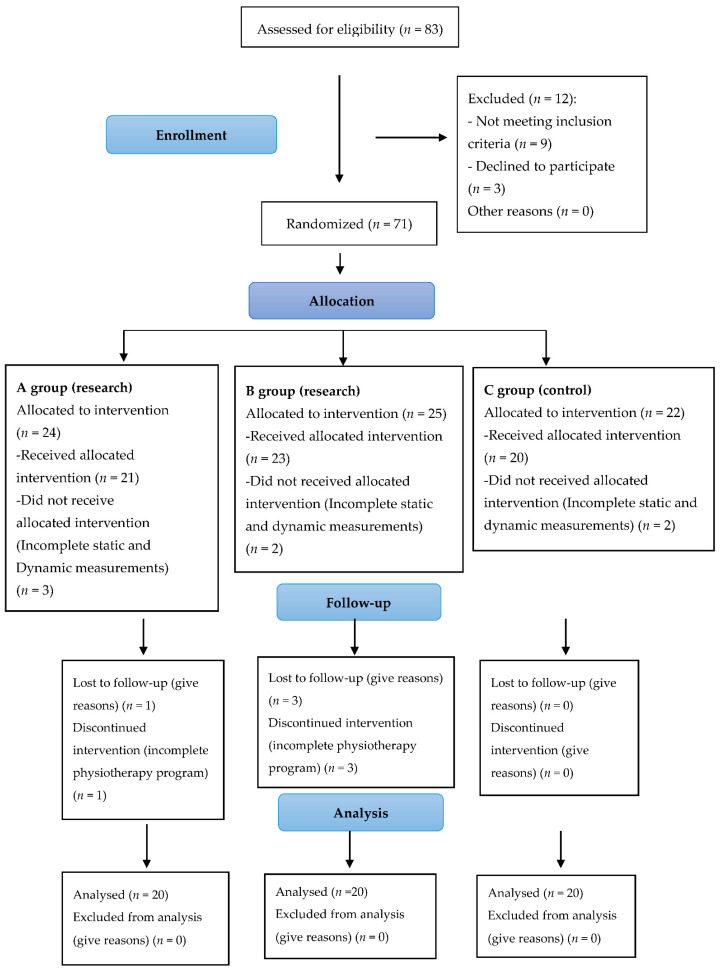
Consort diagram.

**Table 1 ijerph-18-04660-t001:** Morphological characteristics of the studied women.

	Group A				Group B				Group C				*p*
	$\bar{x}$	SD	Min	Max	$\bar{x}$	SD	Min	Max	$\bar{x}$	SD	Min	Max	
Age (years)	65.8	4.4	60	79	64.7	4.5	60	74	67.8	5.0	60	80	0.329
Body height (cm)	156.3	4.4	149	163	159.6	5.4	149	169	157.6	4.8	148	165	0.614
Body weight (kg)	67.4	9.1	54	82	70.5	9.2	55	86	78.7	13.1	61	108	0.737
BMI (kg/m^2)^	27.6	3.4	21	34	27.7	3.5	22	33	31.6	5.1	24	41	0.867

$\bar{x}$—mean, SD—standard deviation, p-level of statistical significance.

**Table 2 ijerph-18-04660-t002:** Differences between selected left foot static indicators in the individual groups before and after therapy.

Left Foot	Group	Before Therapy					After Therapy					Between Measurements Comparison
		$\bar{x}$	Me	Min	Max	SD	$\bar{x}$	Me	Min	Max	SD	
Load (kg)	A	35.35	35.00	23.00	48.00	6.18	35.60	35.00	28.00	47.00	6.26	*p* = 0.717
	B	37.00	36.50	25.00	49.00	6.39	35.90	37.00	26.00	45.00	5.51	*p* = 0.163
	C	40.50	35.50	26.00	83.00	12.50	42.90	36.50	24.00	79.00	15.00	*p* = 0.489
Between groups comparison		A*–*B *p* = 1.000 A*–*C *p* = 0.696 B*–*C *p* = 1.000					A–B *p* = 1.000 A–C *p* = 0.384 B–C *p* = 0.634					
Max. load (gr/cm^2^)	A	917.35	898.00	683.00	1452.00	177.03	930.25	926.50	701.00	1212.00	117.77	*p* = 0.232
	B	868.90	845.00	724.00	1100.00	103.35	898.90	878.50	734.00	1153.00	114.19	*p* = 0.262
	C	969.95	952.50	686.00	1480.00	220.65	927.65	942.00	680.00	1237.00	165.69	*p* = 0.489
Between groups comparison		A*–*B *p* = 1.000 A*–*C *p* = 1.000 B*–*C *p* = 0.489					A–B *p* = 0.831 A–C *p* = 1.000 B–C *p* = 1.000					
Average load (gr/cm^2^)	A	410.20	405.00	315.00	544.00	52.40	432.60	437.50	338.00	538.00	55.03	*p* = 0.019 *
	B	390.85	389.00	295.00	469.00	51.74	408.40	401.00	319.00	511.00	57.36	*p* = 0.022 *
	C	430.40	401.00	283.00	689.00	98.91	422.35	410.50	319.00	635.00	87.77	*p* = 0.856
Between groups comparison		A*–*B *p* = 1.000 A*–*C *p* = 1.000 B*–*C *p* = 0.925					A–B *p* = 0.701 A–C *p* = 0.706 B–C *p* = 1.000					
Forefoot load (%)	A	21.75	23.00	12.00	31.00	5.77	22.95	23.00	14.00	34.00	4.56	*p* = 0.687
	B	27.55	25.50	19.00	43.00	6.61	25.75	25.00	14.00	41.00	7.06	*p* = 0.178
	C	21.40	20.50	11.00	34.00	6.51	23.75	24.00	12.00	38.00	7.11	*p* = 0.071
Between groups comparison		A*–*B *p* = 0.071 A*–*C *p* = 1.000 B*–*C *p* = 0.020 *					A–B *p* = 0.773 A–C *p* = 1.000 B–C *p* = 1.000					
Hindfoot load (%)	A	29.95	29.50	17.00	44.00	6.93	29.60	29.00	15.00	42.00	5.40	*p* = 0.333
	B	25.35	26.50	14.00	41.00	7.94	25.75	26.50	13.00	37.00	6.24	*p* = 0.811
	C	30.00	28.50	15.00	54.00	7.97	29.00	27.00	18.00	43.00	7.36	*p* = 0.704
Between groups comparison		A*–*B *p* = 0.214 A*–*C *p* = 1.000 B*–*C *p* = 0.371					A–B *p* = 0.129 A–C *p* = 1.000 B–C *p* = 0.796					

$\bar{x}$—mean, Me—median, SD—standard deviation, *p*—level of the statistical significance, *—significantly different (*p* < 0.05).

**Table 3 ijerph-18-04660-t003:** Differences between selected right foot static indicators in the individual groups before and after therapy.

Right Foot	Group	Before Therapy					After Therapy					Between Measurements Comparison
		$\bar{x}$	Me	Min	Max	SD	$\bar{x}$	Me	Min	Max	SD	
Load (kg)	A	32.25	31.00	25.00	44.00	5.14	31.80	32.50	25.00	41.00	4.39	*p* = 0.906
	B	33.35	33.00	21.00	49.00	7.09	34.10	33.50	23.00	51.00	7.27	*p* = 0.332
	C	37.50	39.00	11.00	53.00	9.28	37.70	37.00	19.00	58.00	9.60	*p* = 0.728
Between groups comparison		A–B *p* = 1.000 A–C *p* = 0.034 * B–C *p* = 0.139					A–B *p* = 0.997 A–C *p* = 0.039 * B–C *p* = 0.409					
Max. load (gr/cm^2^)	A	924.00	919.00	673.00	1458.00	185.78	927.40	905.00	701.00	1233.00	141.46	*p* = 0.907
	B	858.05	821.50	733.00	1091.00	102.09	889.10	883.50	670.00	1148.00	135.62	*p* = 0.247
	C	916.70	890.50	439.00	1392.00	242.13	882.40	873.50	682.00	1355.00	180.16	*p* = 0.455
Between groups comparison		A–B *p* = 0.761 A–C *p* = 1.000 B–C *p* = 1.000					A–B *p* = 1.000 A–C *p* = 0.779 B–C *p* = 1.000					
Average load (gr/cm^2^)	A	399.30	396.00	306.00	608.00	67.71	411.85	388.00	352.00	518.00	56.25	*p* = 0.185
	B	387.40	376.50	306.00	512.00	56.86	416.05	405.50	251.00	548.00	74.54	*p* = 0.032 *
	C	419.95	411.50	230.00	608.00	92.20	391.10	396.00	259.00	574.00	76.81	*p* = 0.104
Between groups comparison		A–B *p* = 1.000 A–C *p* = 1.000 B–C *p* = 0.610					A–B *p* = 1.000 A–C *p* = 1.000 B–C *p* = 0.675					
Forefoot load (%)	A	19.40	20.00	7.00	30.00	6.30	19.40	20.00	10.00	30.00	5.04	*p* = 1.000
	B	23.15	23.50	10.00	44.00	8.77	23.05	21.50	10.00	40.00	7.33	*p* = 0.948
	C	20.15	20.50	5.000	29.00	6.18	20.90	22.50	5.00	31.00	7.40	*p* = 0.529
Between groups comparison		A–B *p* = 0.639 A–C *p* = 1.000 B–C *p* = 1.000					A–B *p* = 0.388 A–C *p* = 0.825 B–C *p* = 1.000					
Hindfoot load (%)	A	29.00	28.00	17.00	42.00	6.68	28.15	28.00	18.00	40.00	5.98	*p* = 0.522
	B	24.40	25.00	9.00	36.00	7.44	25.45	29.50	13.00	35.00	8.08	*p* = 0.496
	C	28.25	30.00	7.00	42.00	8.47	26.35	27.00	14.00	43.00	7.36	*p* = 0.175
Between groups comparison		A–B *p* = 0.306 A–C *p* = 0.825 B–C *p* = 1.000					A–B *p* = 1.000 A–C *p* = 1.000 B–C *p* = 1.000					

$\bar{x}$—mean, Me—median, SD—standard deviation, *p*—level of the statistical significance, *—significantly different (*p* < 0.05).

**Table 4 ijerph-18-04660-t004:** Differences between selected left foot dynamic indicators in individual groups before and after therapy.

Left Foot	Group	Before Therapy					After Therapy					Between Measurements Comparison
		$\bar{x}$	Me	Min	Max	SD	$\bar{x}$	Me	Min	Max	SD	
Area (cm^2^)	A	108.40	108.50	90.00	139.00	12.45	111.75	109.50	93.00	134.00	11.51	*p* = 0.140
	B	113.65	114.50	85.00	137.00	15.43	109.10	109.00	88.00	134.00	13.04	*p* = 0.021 *
	C	114.85	113.50	89.00	139.00	12.63	112.25	110.00	86.00	146.00	15.58	*p* = 0.478
Between groups comparison		A*–*B *p* = 0.536 A*–*C *p* = 0.268 B*–*C *p* = 1.000					A*–*B *p* = 1.000 A*–*C *p* = 1.000 B*–*C *p* = 1.000					
Max. load (gr/cm^2^)	A	2648.20	2526.00	2100.00	3744.00	441.17	2564.65	2552.00	2315.00	3028.00	192.77	*p* = 0.881
	B	2779.05	2802.00	1952.00	3788.00	461.19	2948.80	2820.00	2192.00	5084.00	717.82	*p* = 0.232
	C	2865.90	2782.00	1764.00	3596.00	457.31	3027.80	2952.00	1956.00	5636.00	728.22	*p* = 0.985
Between groups comparison		A*–*B *p* = 0.964 A*–*C *p* = 0.161 B*–*C *p* = 1.000					A*–*B *p* = 0.091 A*–*C *p* = 0.007 * B*–*C *p* = 1.000					
Average load (gr/cm^2^)	A	945.45	953.50	676.00	1140.00	125.15	937.95	936.00	821.00	1039.00	69.57	*p* = 0.754
	B	1055.10	1030.50	844.00	1280.00	142.98	1094.40	1072.00	857.00	1403.00	169.72	*p* = 0.157
	C	1084.80	1049.50	741.00	1516.00	194.91	1142.90	1163.00	705.00	1458.00	194.87	*p* = 0.147
Between groups comparison		A*–*B *p* = 0.100 A*–*C *p* = 0.04 * B*–*C *p* = 1.000					A*–*B *p* = 0.010 A*–*C *p* < 0.001 * B*–*C *p* = 1.000					
Forefoot load (%)	A	57.35	57.50	42.00	74.00	7.86	60.60	61.00	52.00	69.00	4.83	*p* = 0.098
	B	56.75	57.50	42.00	68.00	5.90	61.75	62.50	40.00	72.00	7.60	*p* = 0.017 *
	C	57.45	58.50	45.00	68.00	7.13	59.25	60.00	47.00	69.00	5.35	*p* = 0.363
Between groups comparison		A*–*B *p* = 1.000 A*–*C *p* = 1.000 B*–*C *p* = 1.000					A*–*B *p* = 1.000 A*–*C *p* = 1.000 B*–*C *p* = 0.442					
Hindfoot load (%)	A	42.65	42.50	26.00	58.00	7.86	39.40	39.00	31.00	48.00	4.83	*p* = 0.098
	B	43.30	42.50	32.00	58.00	5.92	38.25	37.50	28.00	60.00	7.60	*p* = 0.016 *
	C	42.55	41.50	32.00	55.00	7.13	40.75	40.00	31.00	53.00	5.35	*p* = 0.363
Between groups comparison		A*–*B *p* = 1.000 A*–*C *p* = 1.000 B*–*C *p* = 1.000					A*–*B *p* = 1.000 A*–*C *p* = 1.000 B*–*C *p* = 0.442					
Single support (ms)	A	439.70	447.50	352.00	531.00	53.85	448.85	459.00	342.00	528.00	48.31	*p* = 0.455
	B	486.95	483.00	364.00	603.00	68.51	468.35	465.50	351.00	592.00	65.13	*p* < 0.001
	C	490.85	467.50	406.00	695.00	83.5	462.65	456.00	131.00	661.00	112.7	*p* = 0.156
Between groups comparison		A*–*B *p* = 0.094 A*–*C *p* = 0.184 B*–*C *p* = 1.000					A*–*B *p* = 1.000 A*–*C *p* = 1.000 B*–*C *p* = 1.000					
Double support (ms)	A	110.65	107.50	73.00	144.00	16.23	114.95	117.50	48.00	140.00	21.02	*p* = 0.089
	B	125.65	124.50	86.00	163.00	22.27	121.00	121.50	83.00	156.00	20.61	*p* < 0.001
	C	124.00	121.00	88.0	159.00	18.72	125.25	122.00	102.0	169.00	17.74	*p* = 0.681
Between groups comparison		A*–*B *p* = 0.047 * A*–*C *p* = 0.085 B*–*C *p* = 1.000					A*–*B *p* = 1.000 A*–*C *p* = 0.761 B*–*C *p* = 1.000					
Gait cycle (ms)	A	688.65	694.50	548.00	878.00	90.67	687.90	696.00	506.00	918.00	88.79	*p* = 0.985
	B	734.00	726.00	606.00	839.00	64.20	706.35	701.00	545.00	880.00	97.07	*p* = 0.085
	C	719.35	684.00	607.00	985.00	106.2	726.70	695.50	551.00	997.00	119.4	*p* = 0.940
Between groups comparison		A*–*B *p* = 0.234 A*–*C *p* = 1.000 B*–*C *p* = 0.614					A*–*B *p* = 1.000 A*–*C *p* = 1.000 B*–*C *p* = 1.000					

$\bar{x}$—mean, Me—median, SD—standard deviation, *p*—level of the statistical significance, *—significantly different (*p* < 0.05).

**Table 5 ijerph-18-04660-t005:** Differences between selected right foot dynamic indicators in the individual groups before and after therapy.

Right Foot	Group	Before Therapy					After Therapy					Between Measurements Comparison
		$\bar{x}$	Me	Min	Max	SD	$\bar{x}$	Me	Min	Max	SD	
Area (cm^2^)	A	111.85	114.00	63.00	143.00	15.84	113.05	113.00	96.00	139.00	11.48	*p* = 0.672
	B	112.60	113.00	83.00	139.00	14.29	107.20	110.00	88.00	141.00	12.70	*p* = 0.021 *
	C	113.65	112.00	88.00	137.00	13.12	114.40	113.00	90.00	140.00	15.77	*p* = 0.532
Between groups comparison		A–B *p* = 1.000 A–C *p* = 1.000 B–C *p* = 1.000					A–B *p* = 0.595 A–C *p* = 1.000 B–C *p* = 0.342					
Max. load (gr/cm^2^)	A	2475.40	2510.00	1660.00	3160.00	384.45	2621.15	2493.50	2044.00	3824.00	437.25	*p* = 0.156
	B	2651.10	2622.00	1564.00	3348.00	462.62	2915.40	2812.00	2368.00	4180.00	458.33	*p* = 0.034 *
	C	2901.90	2648.00	2100.00	5408.00	743.20	2703.80	2724.00	2204.00	3500.00	380.25	*p* = 0.313
Between groups comparison		A–B *p* = 0.465 A–C *p* = 0.088 B–C *p* = 1.000					A–B *p* = 0.087 A–C *p* = 1.000 B–C *p* = 0.361					
Average load (gr/cm^2^)	A	941.65	931.50	703.00	1214.00	123.73	920.10	931.50	710.00	1169.00	111.13	*p* = 0.415
	B	1004.8	1002.0	704.00	1288.00	150.56	1109.3	1132.5	776.00	1422.00	175.41	*p* = 0.006 *
	C	1075.3	1017.0	822.00	1512.00	196.80	1073.7	1062.0	753.00	1434.00	198.94	*p* = 0.970
Between groups comparison		A–B *p* = 0.519 A–C *p* = 0.068 B–C *p* = 1.000					A–B *p* = 0.001 * A–C *p* = 0.029 * B–C *p* = 1.000					
Forefoot load (%)	A	55.75	54.00	43.00	79.00	8.82	60.85	63.00	49.00	69.00	6.62	*p* = 0.011 *
	B	52.75	53.00	31.00	66.00	10.30	59.85	61.00	46.00	69.00	6.70	*p* = 0.002 *
	C	57.15	57.50	43.00	70.00	8.09	60.50	60.00	50.00	72.00	5.78	*p* = 0.054
Between groups comparison		A–B *p* = 1.000 A–C *p* = 1.000 B–C *p* = 0.581					A–B *p* = 1.000 A–C *p* = 1.000 B–C *p* = 1.000					
Hindfoot load (%)	A	44.25	46.00	21.00	57.00	8.82	38.65	36.50	27.00	51.00	7.15	*p* = 0.009 *
	B	47.25	47.00	34.00	69.00	10.30	40.15	39.00	31.00	54.00	6.70	*p* = 0.003 *
	C	42.85	42.50	30.00	57.00	8.09	39.50	40.00	28.00	50.00	5.78	*p* = 0.054
Between groups comparison		A–B *p* = 1.000 A–C *p* = 1.000 B–C *p* = 0.581					A–B *p* = 1.000 A–C *p* = 1.000 B–C *p* = 1.000					
Single support (ms)	A	485.65	455.00	353.00	895.00	122.45	475.20	446.00	346.00	747.00	93.07	*p* = 0.765
	B	479.05	477.50	374.00	597.00	57.795	462.00	455.00	361.00	605.00	59.986	*p* < 0.001 *
	C	489.15	490.50	388.00	644.00	74.217	475.80	486.50	253.00	629.00	94.154	*p* = 0.444
Between groups comparison		A–B *p* = 1.000 A–C *p* = 0.756 B–C *p* = 1.000					A–B *p* = 1.000 A–C *p* = 1.000 B–C *p* = 1.000					
Double support (ms)	A	125.45	122.00	86.00	175.0	23.17	127.00	121.50	94.00	191.0	21.89	*p* = 0.720
	B	125.65	122.00	104.00	154.0	16.339	120.80	120.50	93.00	147.0	15.900	*p* < 0.001 *
	C	126.35	127.50	97.00	156.00	16.20	127.55	127.00	96.00	174.00	22.05	*p* = 0.730
Between groups comparison		A–B *p* = 1.000 A–C *p* = 1.000 B–C *p* = 1.000					A–B *p* = 1.000 A–C *p* = 1.000 B–C *p* = 0.957					
Gait cycle (ms)	A	718.95	688.50	540.00	957.0	117.16	727.95	691.50	533.00	1114.0	133.94	*p* = 0.822
	B	753.05	739.50	621.00	1061.0	97.246	700.45	697.50	547.00	913.0	91.158	*p* = 0.009 *
	C	717.45	719.50	587.00	933.00	89.31	689.95	717.00	49.00	978.00	187.06	*p* = 0.910
Between groups comparison		A–B *p* = 0.675 A–C *p* = 1.000 B–C *p* = 0.874					A–B *p* = 1.000 A–C *p* = 1.000 B–C *p* = 1.000					

$\bar{x}$—mean, Me—median, SD—standard deviation, *p*—level of the statistical significance, *—significantly different (*p* < 0.05).

**Table 6 ijerph-18-04660-t006:** Comparison of the balance test results in the studied groups before and after therapy.

Variable	Group	Before Therapy					After Therapy					Between Measurements Comparison
		$\bar{x}$	Me	Min	Max	SD	$\bar{x}$	Me	Min	Max	SD	
Postural Stability Test (PST)	A	0.52	0.50	0.30	1.00	0.18	0.49	0.40	0.30	0.90	0.16	*p* = 0.489
	B	0.55	0.50	0.30	1.00	0.22	0.50	0.45	0.20	1.30	0.26	*p* = 0.272
	C	0.58	0.50	0.20	1.10	0.23	0.61	0.60	0.30	1.50	0.26	*p* = 0.281
Between groups comparison		A*–*B *p* = 1.000 A*–*C *p* = 1.000 B*–*C *p* = 1.000					A*–*B *p* = 1.00 A*–*C *p* = 0.285 B*–*C *p* = 0.446					
Limits of Stability (LOS %)	A	40.70	42.00	16.00	62.00	12.00	47.25	45.00	23.00	69.00	12.04	*p* = 0.415
	B	43.25	42.00	26.00	69.00	10.52	32.60	31.50	7.00	61.00	12.98	*p* = 0.005 *
	C	39.45	41.50	21.00	54.00	9.90	34.35	35.00	20.00	49.00	8.00	*p* = 0.480
Between groups comparison		A*–*B *p* = 0.939 A*–*C *p* = 0.082 B*–*C *p* = 0.162					A*–*B *p* = 0.443 A*–*C *p* = 0.023 * B*–*C *p* < 0.001*					
Eyes open hard ground	A	0.73	0.63	0.50	1.78	0.30	0.58	0.58	0.37	1.12	0.17	*p* = 0.151
	B	0.72	0.69	0.39	1.10	0.20	0.68	0.61	0.32	1.19	0.26	*p* = 0.006 *
	C	0.72	0.73	0.41	1.17	0.22	0.74	0.63	0.39	1.41	0.26	*p* = 0.765
Between groups comparison		A*–*B *p* = 0.651 A*–*C *p* = 0.651 B*–*C *p* < 0.001 *					A*–*B *p* = 0.998 A*–*C *p* = 0.122 B*–*C *p* = 0.843					
Eyes closed hard ground	A	0.83	0.80	0.41	1.87	0.35	0.79	0.72	0.39	1.79	0.30	*p* = 0.261
	B	0.76	0.69	0.37	1.63	0.28	0.76	0.79	0.29	1.56	0.29	*p* = 0.075
	C	0.84	0.82	0.45	1.23	0.20	0.88	0.81	0.46	1.89	0.32	*p* = 0.845
Between groups comparison		A*–*B *p* = 1.000 A*–*C *p* = 1.000 B*–*C *p* = 0.409					A*–*B *p* = 1.000 A*–*C *p* = 0.906 B*–*C *p* = 0.831					
Eyes open soft ground	A	1.44	1.38	1.00	1.88	0.22	1.11	1.13	0.63	1.51	0.22	*p* < 0.001 *
	B	1.44	1.37	1.01	1.91	0.31	1.25	1.20	0.91	1.76	0.26	*p* = 0.003 *
	C	1.58	1.56	0.96	2.93	0.43	1.56	1.48	0.80	2.51	0.50	*p* = 0.881
Between groups comparison		A*–*B *p* = 1.000 A*–*C *p* = 1.000 B*–*C *p* = 0.874					A*–*B *p* = 0.554 A*–*C *p* = 0.005 * B*–*C *p* = 0.208					
Eyes closed hard ground	A	2.81	2.78	1.96	3.59	0.42	2.61	2.63	1.73	3.49	0.50	*p* = 0.094
	B	2.63	2.70	1.69	3.29	0.40	2.40	2.43	1.69	3.32	0.49	*p* = 0.068
	C	2.93	2.79	1.74	4.24	0.72	2.67	2.89	1.77	3.71	0.59	*p* = 0.108
Between groups comparison		A*–*B *p* = 0.576 A*–*C *p* = 1.000 B*–*C *p* = 0.510					A*–*B *p* = 0.554 A*–*C *p* = 1.000 B*–*C *p* = 0.342					

$\bar{x}$—mean, Me—median, SD—standard deviation, *p*—level of the statistical significance, *—significantly different (*p* < 0.05).

## Data Availability

The data presented in this study are available on request from the corresponding author.

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
