# Peer review of "The Influence of the Physiotherapeutic Program on Selected Static and Dynamic Foot Indicators and the Balance of Elderly Women Depending on the Ground Stability"

_ijerph, 2021, doi:10.3390/ijerph18094660_

Round 1

Reviewer 1 Report

First, I would like to congratulate you for adding new results related to fundamental aspects to be taken into account for an adequate physical exercise program for the elderly and, on the other hand, thank you for the opportunity to review the manuscript. I hope and wish the best and for any question, always available.

Best regards.

Author Response

Response to Reviewer 1

Dear Reviewer,

We would like to thank you for accepting our manuscript.

Reviewer 2 Report

The aim of this paper was to assess a rehabilitation program effects' in old population, focusing on balance strategy in static/dynamic conditions. Methodology used was good, this was a three groups randomized study with good statistic relevance described. My only suggestion is to verify English quality, (i'm not a native English speaker) to improve quality of the paper. 

Author Response

Response to Reviewer 2

Dear Reviewer,

We would like to thank you for accepting our manuscript and explain that our manuscript was revised after translation by native speaker.

Reviewer 3 Report

Reviewer’s comments 

This is a study aimed to assess the impact of the authors’ physiotherapeutic programme on selected indicators of motor functions of elderly women depending on the ground stability.

The topic of physical exercises care in ground stability is important for the elderly, especially because of the high incidence and prevalence of falls in the elderly. However, this study should be reviewed for better use in the clinical  and scientific population.

 TITLE AND OBJECTIVE

Reviewer: It is unnecessary to include the word "Authors" in the title and objective. This information would be better positioned in the methodology part.

ABSTRACT

Line 13:

Reviewer: The objective is not clear the need to check about the influence programme conducted on unstable ground. This topic would be more specific to direct the conclusion. Here and at the end of the introduction.

This seems to be the differential of the study. Value that fact.

Line 20:

Reviewer: The "static test" information is not clear when it is associated with "balance assessment".

Line 23:

“The greatest number of statistically significant changes was obtained in the group exercising on unstable ground.”

Reviewer: What do you  conclude with this information?

INTRODUCTION

This topics there are not specific hypotheses being tested,  controversial and divergent hypotheses and  the main objective at the end of the introduction, as pointed out in the instructions to authors

METHODS 

Line 175: “position of her feet”

Reviewer: What was the position?

Lines 196 and 202: “low-, half-high- and high-position exercises.

Reviewer: This frase is not clear.

- What the analyzed variables can bring ​clinical information to benefit the studied population.

- Detail the type of exercise, resource and which muscle groups were worked.

DATA ANALYSIS

Reviewer:

- I think that two way ANOVA test would be enough to answer the objective of the study.

- I did not find which values were used for statistic analysis. For example, mean value.

RESULTS

Reviewer:

- Identify the tests used in the legend of each table or in a common initial text for all results.

- I did not identify results performed by anova test.

- The results are not conclusive and this is difficult to do the discussion. I suggest reviewing the statistical test.

DISCUSSION

Reviewer:

- In the discussion, there is a lot content that could have been placed in the introduction.

CONCLUSION

The results showed values that do not allow concluding these results. A rigorous discussion is necessary to associate the results with the conclusions.

Author Response

Response to Reviewer 3

Dear Reviewer,

We would like to thank you for a detailed review of our manuscript and all of your valuable remarks. We have addressed them all in detail below.

Title and objective

  • We removed word „Authors`’ form the title due to Reviewer`s suggestion

Abstract

  • Line 13 – we corrected the sentence due to Reviewer`s suggestion
  • Line 20 – we added word „foot’ to clarify the sentence due to Reviewer`s suggestion
  • Line 23 – we corrected the sentence due to Reviewr`s suggestion

Introduction

  • We added specific hypothesis in the Introduction section and we verified it in the Discussion section due to Reviewer`s suggestion

Methods

  • Line 175 – we explained the sentence due to Reviewer`s suggestion
  • Line 196 and 202 – we corrected words due to Reviewer`s suggestion
  • We added practical conclusion in the Conclusion section due to Reviewer`s suggestion
  • We completed information due to Reviewer`s suggestion

Data analysis

  • The two-way ANOVA test was not used for the statistical analysis, because the test requires meeting certain assumptions:
    • the normality of distribution of variables in individual groups or the normality of distribution of differences between the variables if the variables from consecutive measurements in the same group are compared
    • homogeneity of variance

As these conditions were not met for every variable, where it was possible to compare inter and group intra parametric tests (t-Student and ANOVA) were used, in other cases their non-parametric equivalents were used.

  • As already mentioned, some of the data in individual groups did not have a normal distribution, therefore, in Tables 2 and following, both the mean and the median are presented.The proper representation of the average value in the group where the variable had a normal distribution is the mean, so where there was no normal distribution, the group is better described by the median.

Results

  • As descriptive statistics as well as inter and intra group comparisons are combined in the table, the results of all applied statistical tests are included in each table.
  • The ANOVA test results are included in a line signed as “between group comparison”
  • The statistical analysis was performed by a professional statistician.We understand that the results are not conclusive, but this is very rare in articles about rehabilitation programs

Discussion

  • We moved the paragraph from the Discuusion section to the Introduction section due to Reviewer`s suggestion. At the same time, we wanted to ask the Reviewer to agree to keep the current shape of the discussion, as it received recognition of the other two reviewers.

Conclusion

  • We corrected the conclusion due to Reviewer`s suggestion

Round 2

Reviewer 3 Report

After doing a detailed reading of the study, I realized that the points were revised or justified. The study presents solid scientific information for publication, therefore, I suggest another way to write the objective.
Some points have been suggested for polishing in this article, but it is a suggestion of writing style, for example, discussion comments made in the first review.

ABSTRACT
Line 13
Reviewer: I suggest "The aim of this study was to assess the impact of the physiotherapeutic programme conducted on unstable ground on selected indicators of motor functions of elderly women" because the stable ground is a reference for analysis.